# Unperturbed expression bias of imprinted genes in schizophrenia

Attila Gulyás-Kovács[1,2], Ifat Keydar[1,2,8], Eva Xia[1,3], Menachem Fromer[2,4,9], Gabriel Hoffman [2], Douglas Ruderfer[2,4,10], CommonMind Consortium, Ravi Sachidanandam [5] & Andrew Chess[1,2,6,7]

How gene expression correlates with schizophrenia across individuals is beginning to be examined through analyses of RNA-seq from postmortem brains of individuals with disease and control brains. Here we focus on variation in allele-specific expression, following up on the CommonMind Consortium (CMC) RNA-seq experiments of nearly 600 human dorso-lateral prefrontal cortex (DLPFC) samples. Analyzing the extent of allelic expression bias—a hallmark of imprinting—we find that the number of imprinted human genes is consistent with lower estimates (≈0.5% of all genes), and thus contradicts much higher estimates. Moreover, the handful of putatively imprinted genes are all in close genomic proximity to known imprinted genes. Joint analysis of the imprinted genes across hundreds of individuals allowed us to establish how allelic bias depends on various factors. We find that age and genetic ancestry have gene-specific, differential effect on allelic bias. In contrast, allelic bias appears to be independent of schizophrenia.

[1] Department of Cell, Developmental and Regenerative Biology, ISMMS, New York, NY 10029, USA. [2] Institute for Genomics and Multiscale Biology, Department of Genetics and Genomic Sciences, ISMMS, New York, NY 10029, USA. [3] Neuroscience Program, The Graduate School of Biomedical Sciences, ISMMS, New York, NY 10029, USA. [4] Division of Psychiatric Genomics, Department of Psychiatry, ISMMS, New York, NY 10029, USA. [5] Department of Oncological Sciences, ISMMS, New York, NY 10029, USA. [6] Fishberg Department of Neuroscience, ISMMS, New York, NY 10029, USA. [7] Friedman Brain Institute, ISMMS, New York, NY 10029, USA. [8] Present address: The Simon And Katya Michaeli Bioinformatics Laboratory For The Research Of The Genome Department of Human Molecular Genetics & Biochemistry, Sackler Medical School, Tel Aviv University, Tel Aviv-Yafo 69978, Israel. [9] Present address: Verily Life Sciences, 94080 South San Francisco, USA. [10] Present address: Division of Genetic Medicine, Departments of Medicine, Psychiatry and Biomedical Informatics, Vanderbilt University, Nashville, TN 37235, USA. These authors contributed equally: Attila Gulyás-Kovács, Ifat Keydar. A full list of consortium members appears at the end of the paper. Correspondence and requests for materials should be addressed to A.C.(email: andrew.chess@mssm.edu)

The observation[1,2] that maternally derived microduplications at 15q11-q13—harboring the imprinted gene UBE3A—may not only cause Prader-Willi syndrome, but also are highly penetrant for schizophrenia (SCZ) has raised the possibility that perturbation of regulation of imprinted genes in general may play a role in psychotic disorders. As it is known that the extent of imprinting of individual genes varies over different tissues, we chose to analyze the dorsolateral prefrontal cortex (DLPFC) region, which controls complex cognitive and executive functions, and is known to display functional abnormalities in SCZ.

A related question is the number of imprinted genes in the human brain. Some 1300 genes were estimated to be imprinted in the mouse brain[3], but followup studies using mouse or human subjects arrived at estimates that are lower with an order of magnitude[4–7,9].

We obtained DLPFC RNA-seq data from the CMC[8] (http://www.synapse.org/CMC) and analyzed allele-specific expression with the idea of (i) identifying imprinted genes in the adult human brain and (ii) explaining the variability in allelic bias across 579 individuals in terms of their psychiatric diagnosis, age at death, etc. This was facilitated by the balanced case–control groups (258 SCZ, 267 control, 54 bipolar, or other affective/mood disorder, AFF) and the large age variability in the cohort.

## Results

**Identification of imprinted genes in the adult human brain.** For each individual $i$ and gene $g$ we quantified allelic bias based on RNA-seq reads using a statistic called read count ratio $S_{ig}$ (Fig. 1, Methods), which ranges from 0.5 to 1 indicating unbiased biallelic expression (at 0.5), some allelic bias (at intermediate values), or strictly monoallelic expression (at 1). We corrected for a number of factors this approach is known to be sensitive to. We quality-filtered RNA-seq reads, and helped distinguish allele-specific reads using DNA genotyping data before calculating $S$ and then applied post hoc corrections for mapping bias ("Methods").

Of 15,584 genes with RNA-seq data 5307 genes passed our filters designed to remove genes with scarce RNA-seq data reflecting low expression and/or low coverage of RNA-seq (Methods: "Quality filtering"). Figure 2 presents the conditional empirical distribution of $S_g$ across all individuals given each gene $g$.

The observed wide $S_g$ distributions suggest large across-individuals variation of allelic bias for all genes, even if a substantial component of the $S_g$ variation originates from technical sources. Still, as expected, for many genes known to be imprinted in mice or in other human tissues (referred to as known imprinted genes like PEG10, ZNF331) the distribution of $S_g$ was shifted to the right signaling strong allelic bias (Fig. 2, upper half).

To identify imprinted genes in the human adult DLPFC, we defined the score of each gene $g$ as the fraction of individuals $i$ for whom $S_g > 0.9$. We ranked all 5307 genes according to their score (Fig. 2 bottom right). An alternative definition of the score, $S_g > 0.7$, yielded similar ranking (Supplementary Fig. 11). The heat map of the $S_g$ distribution for ranked genes (Fig. 2, lower left) shows that the top 50 genes, which constitute ≈1% of all genes in our analysis, are qualitatively different from the bottom ≈99% exhibiting strongly right-shifted distribution of $S_g$ characteristic to imprinting.

The 29 of the top-scoring 50 genes fell into previously described imprinted gene clusters (Supplementary Fig. 1); 21 of these 29 are known imprinted genes, while 8 are nearby candidates defined as genes near (<1 Mb of) known imprinted ones but themselves previously not shown to be imprinted (blue and green $y$-axis labels in Fig. 3). A priori the expectation is that known imprinted genes and nearby candidates are much more likely to be imprinted in the present data set than distant candidates defined as genes that neither belong to nor localize near known imprinted genes (Fig. 3, red $y$-axis labels). We combined this prior expectation with two tests based on our data to distinguish imprinting from alternative causes of high read count ratio such as mapping bias and cis-eQTL effects[9] (see Methods: "Reference/nonreference allele test" and "Test for nearly unbiased allelic expression"). The results of both tests ($X$'s and black bars, Fig. 3) agreed well with the a priori expected status. This prompted us to call imprinted in the adult human DLPFC those genes in the top 50 that are either known imprinted or nearby candidates. We included also the known imprinted gene UBE3A, which ranked below 50 but whose score was still substantial (Supplementary Fig. 2), yielding 30 imprinted genes (panel headers in Figs. 4–5).

**Explaining the variability in allelic bias of imprinted genes.** Getting at the central question of our work Fig. 4 shows that read

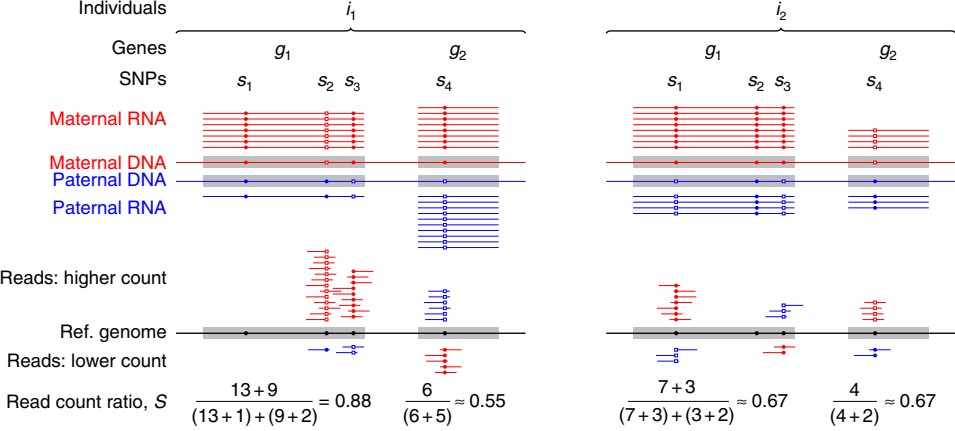

**Fig. 1** Quantifying allelic bias of expression in human individuals using the RNA-seq read count ratio statistic $S_{ig}$. The strength of bias towards the expression of the maternal (red) or paternal (blue) allele of a given gene $g$ in individual $i$ is gauged with the count of RNA-seq reads carrying the reference allele (small closed circles), and the count of reads carrying an alternative allele (open squares) at all SNPs for which the individual is heterozygous. The allele with the higher read count tends to match the allele with the higher expression, but measurement errors may occasionally revert this tendency as seen for SNP $s_3$ in gene $g_1$ in individual $i_2$

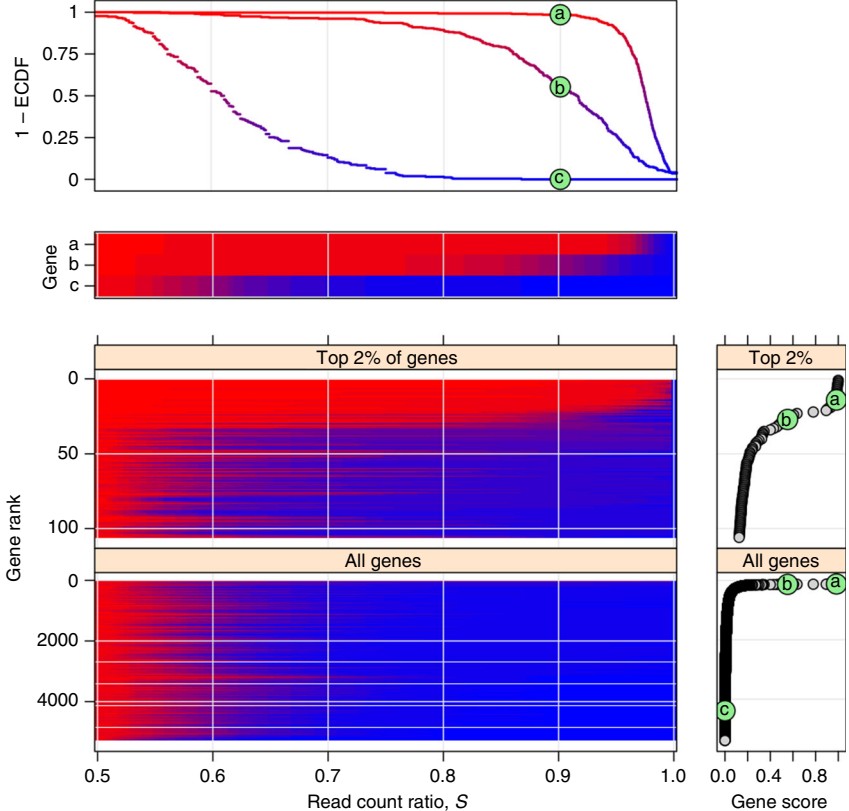

**Fig. 2** Across-individuals distribution of read count ratio $S$ for each gene indicates substantial variation of allelic bias and that <1% of all genes are imprinted. The vertically arranged four main panels present the empirical distribution of $S_g$ across all individuals given each gene $g$. The upper two panels are distinct representations (survival plot: 1 – ECDF, and "survival heatmap") of the same three distributions corresponding to a: PEG10, b: ZNF331, and c: AFAP1. PEG10 and ZNF331, previously found to be imprinted in mice or in other human tissues, and one for AFAP1, a gene without prior evidence. The bottom two survival heatmaps present the distribution of $S_g$ for the top 2% and 100% of the 5307 analyzed genes. These are ranked according to gene score defined as 1 – ECDF (0.9) in the bottom far right panels. The score of PEG10, ZNF331, and AFAP1 is marked by a,b,c, respectively, in green circles. As expected, PEG10 and ZNF331 both score high and rank within the top 30 of all genes suggesting they are also imprinted in the present context, the adult human DLFPC. The bottom panels also indicate that <1% of all genes might be imprinted

count ratio is similarly distributed in the control, SCZ, and AFF group for all 30 imprinted genes, suggesting independence between allelic bias and diagnosis of SCZ. Similar pattern was observed for not imprinted genes (Supplementary Fig. 12).

To support the above qualitative result on imprinted genes, we fitted several fixed and mixed effects models[10] that model the dependence of read count ratio jointly on all explanatory variables (Methods: "Statistical models-informal overview," and beyond). Such joint models can capture much of the complex pattern of dependencies in genomic data, including those we observed within and between technical and biological explanatory variables (Supplementary Table 1, Supplementary Fig. 3). For both the fixed and mixed class, we selected the model that fitted the data the best (unlm.Q/wnlm.Q for both fixed and mixed models, Supplementary Figs. 6-8). Fixed and mixed models also agreed qualitatively on gene-specific coefficients reporting effects/ dependencies (Supplementary Fig. 9-10). We based final inference on the selected mixed model because that gains power from letting genes "borrow strength from each other" (Supplementary Fig. 5).

Based on the best fitting mixed model (henceforth "the model") we could formally reject the hypotheses that read count ratio depends on diagnosis as either main effect or interaction (see term (1|Dx) and (1|Dx):Gene in Table 1, respectively). This key result is not due to low power. This is because in the mixed model

the gene variable (which identifies the gene that a particular data point corresponds to) is similar to the Dx variable (reporting on disease status) in that they are both categorical and are modeled as random effects. If Dx had an effect size that is comparable to the effect of gene then that effect would be detected by our model-based inference, since the effect of gene is highly significant. See (1|Gene) in Table 1 and compare panels in Fig. 4).

Scatter plots suggested that the read count ratio depends negatively on age for some imprinted genes, depends positively for others, and is independent of age for the rest of imprinted genes (Fig. 5 and Supplementary Fig. 4). This apparent dependence might be indirect, i.e., one that is mediated by some variable(s) "inbetween" age and read count ratio (Supplementary Figs. 3, 5), but the model allowed us to isolate the direct component of age dependence: we found that the gene-specific random age effect is indeed significant even if no fixed effect— which would be shared by all imprinted genes—was supported (see (Age|Gene) and age, respectively, in Table 1).

Based on the model we also predicted gene-specific regression coefficients mediating the direct component of age effect (Supplementary Fig. 10 top middle). The predicted coefficients agreed well with all but a few panels of Fig. 5 the latter of which (e.g., UBE3A) therefore represent purely indirect dependence.

The same type of analysis on the effects of ancestry principal components and gender gave similar results: while the fixed

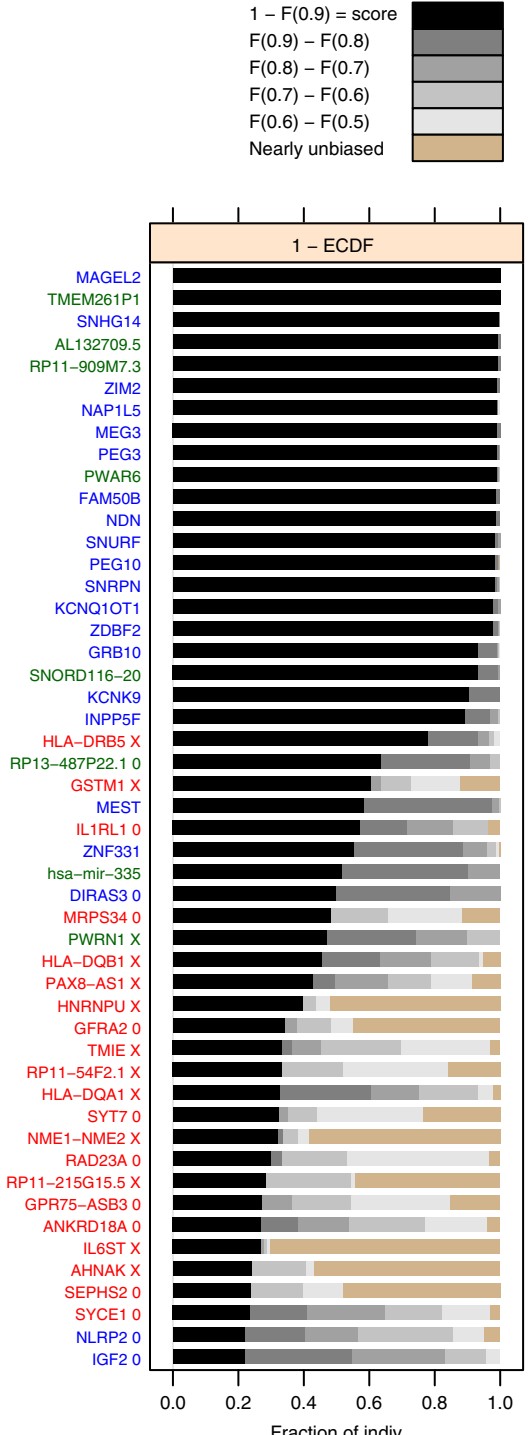

**Fig. 3** The top 50 genes ranked by the gene score defined, for gene $g$, as $1 - F_g(0.9)$, where $F_g$ is the empirical cumulative distribution function (ECDF) for $g$. Thus $1 - F_g(0.9)$, is the fraction of individuals $i$ for which $S_{ig} > 0.9$. Note that the same ranking and score is shown in the bottom half of Fig. 2. The tan colored bars indicate the fraction of individuals with nearly unbiased expression (Eq. 2). Gene names ($y$ axis) are colored according to prior imprinting status: known imprinted (blue), nearby candidate (green), and distant candidate (red). "X" characters next to gene names indicate mapping bias and/or cis-eQTL effects based on the reference/nonreference allele test ("Methods"), while "0" indicates that total allele count was insufficient for this test

effect, shared by all genes, of these variables was negligible, three of the random, gene-specific, effects received significant support. These three, ordered by decreasing statistical significance, are (Ancestry.1|Gene), (Ancestry.3|Gene), and (1|Gender:Gene) (Table 1). The corresponding predicted random coefficients are presented in Supplementary Fig. 10.

Although some of the random effects described above are statistically significant (Table 1), their size is relatively subtle (Fig. 5). Nonetheless, we focussed on genes with very strong allelic bias, in other words nearly monoallelic expression, because that is a hallmark of imprinted genes. In these genes even a subtle change in allelic bias may lead to a qualitative change that carries some biological significance.

Finally, we fitted the same mixed models to two subsets of data, each containing only 15 genes. The results (Supplementary Table 3) are qualitatively similar to those based on the full data set with 30 genes apart from a few marked differences that are explained by the reduction in both the number of data points and in the variability of certain effects across genes.

In summary age, ancestry, and to a lesser extent gender, are suggested by our model-based analysis to exert effect on allelic bias in a way that the direction and magnitude of the effect varies across genes.

## Discussion

The number of imprinted genes in the mammalian brain has been controversial: some early genome wide studies[3,11] estimated over a thousand, suggesting that the number of imprinted genes in the brain is an order of magnitude greater than in other tissues. Later work cast doubt on the methodology used and found that the number of imprinted genes in brain is in line with expectations from studies of other tissues, identifying only a handful of new candidate imprinted genes in brain[5–7]. Based on 579 postmortem human DLPFC samples, we find evidence supporting only a handful of novel imprinted genes all of which reside in genomic locations nearby to known imprinted genes. Thus our results support those more recent studies that found no large excess of imprinted genes in the brain.

The large size of our sample and the case–control makeup allowed us to explore the potential for correlation of extent of imprinting in the DLPFC with SCZ. Although our approach gave strong support for dependence of imprinting on age and ancestry, no dependence on SCZ was detected either when we assumed that the dependence is the same for all imprinted genes or that it varies across genes. Thus our data indicate that imprinting in the DLFPC does not play a significant role in SCZ in contradiction of the "imprinted brain" hypothesis[12]. Given the complex genetic architecture of SCZ[13] as well as technical noise in postmortem brain RNA studies there could still be some correlation of the extent of imprinting and SCZ.

We found that imprinting depends on ancestry in a gene-specific manner, but the type of dependence that is shared by all imprinted genes was not supported. This is expected because the studied ancestry variables must incorporate some of the cis expression QTLs in imprinted genes such that those eQTLS perturb allelic bias in a gene-specific manner.

Our finding that imprinting depends on age in later adulthood is rather intriguing given the quantitative relationship between epigenetics and aging[14]. Age dependence of imprinting through early postnatal life supported experimentally[7], but such dependence during later adulthood has so far only been predicted[15] based on a hypothesis that links "genomic imprinting and the social brain"[16]. Previous genomics studies[5] were statistically underpowered to address this question in humans. Although our age-related finding supports the "social brain" hypothesis, it leaves the possibility open

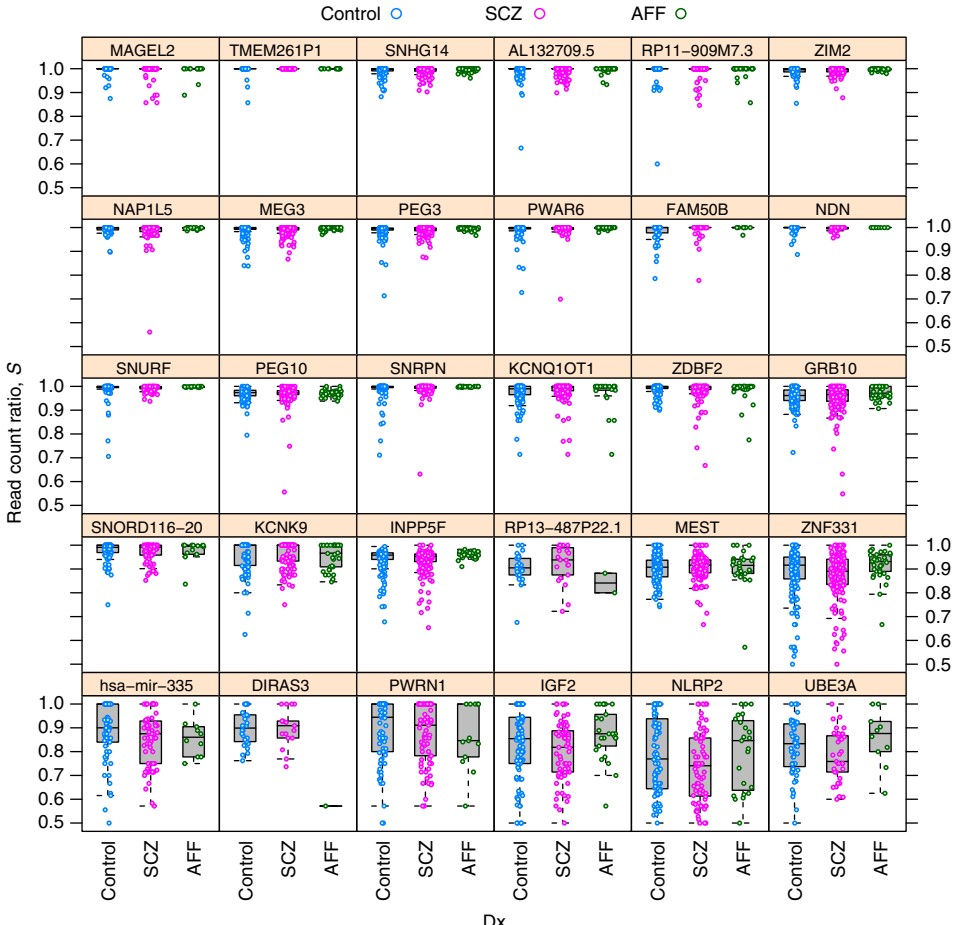

**Fig. 4** Schizophrenia does not affect allelic bias of imprinted genes. Distribution of read count ratio for control, schizophrenic (SCZ), and affectic spectrum (AFF) individuals within each gene that has been considered as imprinted in the DLPFC brain area in this study

that the observed age-related changes indicate merely the loss of tight regulation of those genes with aging.

## Methods

**Defining the read count ratio to quantify allelic bias**. We quantified allelic bias based on RNA-seq reads using a statistic called read count ratio $S$, whose definition we based on the total read count $T$ and the higher read count $H$, i.e., the count of reads carrying only either the reference or the alternative SNP variant, whichever is higher. The definition is

$$S_{ig} = \frac{H_{ig}}{T_{ig}} = \frac{\sum_s H_s}{\sum_s T_s}, \tag{1}$$

where $i$ identifies an individual, $g$ a gene, and the summation runs over all SNPs $s$ for which gene $g$ is heterozygous in individual $i$ (Fig. 1). Note that if $B_{ig}$ is the count or reads that map to the $b_{ig}$ allele (defined as above) and if we make the same distributional assumption as above, namely that $B_{ig} \sim \text{Binom}(p_{ig}, T_{ig})$, then $\Pr(H_{ig} = B_{ig}|p_{ig})$, the probability of correctly assigning the reads with the higher count to the allele toward which expression is biased, tends to 1 as $p_{ig} \rightarrow 1$. We took advantage of this theoretical result in that we subjected only those genes to statistical inference, whose read count ratio was found to be high and, therefore, whose $p_{ig}$ is expected to be high as well.

Figure 1 illustrates the calculation of $S_{ig}$ for the combination of two hypothetical genes, $g_1, g_2$, and two individuals, $i_1, i_2$. It also shows an example for the less likely event that the lower rather than the higher read count corresponds to the SNP variant tagging the higher expressed allele (see SNP $s_3$ in gene $g_1$ in individual $i_2$).

Before we carried out our read count ratio-based analyses, however, we cleaned our RNA-seq data by quality-filtering and by improving the accuracy of SNP calling with the use of DNA SNP array data and imputation. In the following subsections of "Methods" we describe the data, these procedures, as well as our regression models in detail.

**Brain samples, RNA-seq**. Human RNA samples were collected from the DLPFC of the CommonMind consortium from a total of 579 individuals after quality control. Subjects included 267 control individuals, as well as 258 with SCZ, and 54 with affective spectrum disorder (AFF). RNA-seq library preparation uses Ribo-Zero (which selects against ribosomal RNA) to prepare the RNA, followed by Illumina paired end library generation. RNA-seq was performed on Illumina HiSeq 2000.

**Mapping, SNP calling and filtering**. We mapped 100 bp, paired-end RNA-seq reads (≈50 million reads per sample) using Tophat to Ensembl gene transcripts of the human genome (hg19; February, 2009) with default parameters and 6 mismatches allowed per pair (200 bp total). We required both reads in a pair to be successfully mapped, and we removed reads that mapped to >1 genomic locus. Then, we removed PCR replicates using the Samtools rmdup utility; around one-third of the reads mapped (which is expected, given the parameters we used and the known high repeat content of the human genome). We used Cufflinks to determine gene expression of Ensembl genes, using default parameters. Using the BCFtools utility of Samtools, we called SNPs (SNVs only, no indels). Then, we invoked a quality filter requiring a Phred score >20 (corresponding to a probability for an incorrect SNP call <0.01).

We annotated known SNPs using dbSNP (dbSNP 138, October 2013). Considering all 579 samples, we find 936,193 SNPs in total, 563,427 (60%) of which are novel. Further filtering of this SNP list removed the novel SNPs and removed SNPs that either did not match the alleles reported in dbSNP or had more than 2 alleles in dbSNP. We also removed SNPs without at least 10 mapped reads in at least one sample. Read depth was measured using the Samtools Pileup utility. After these filters were applied, 364,509 SNPs remained in 22,254 genes. These filters enabled use of data with low coverage. For the 579 samples, there were 203 million reads overlapping one of the 364,509 SNPs defined above. Of those, 158 million (78%) had genotype data available from either SNP array or imputation.

**Genotyping and calibration of imputed SNPs**. DNA samples were genotyped using the Illumina Infinium SNP array. We used PLINK with default parameters to

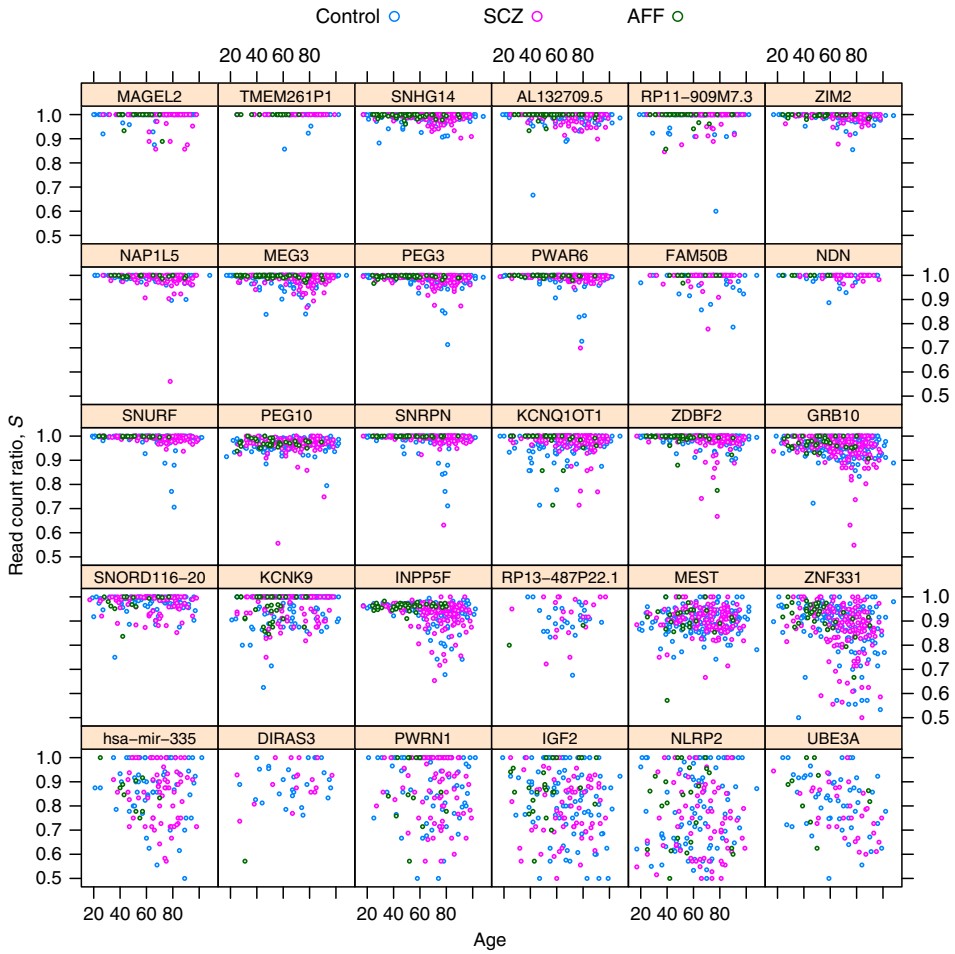

**Fig. 5** Allelic bias depends differentially on age across imprinted genes. The panels and colors are consistent with the imprinted genes and psychiatric diagnoses presented in Fig. 4. The differential dependence on age is apparent when comparing PEG3 or ZNF331 (negative dependence) to KCNK9 or RP13-487P22.1 (positive dependence) or to NDN or NLRP2 (no dependence)

**Table 1 Hypothesis tests concerning the effect of various predictor terms on the quasi log-transformed read count ratio $Q$ of imprinted genes, interpreted as effects on allelic bias**

| Hypothesis | | Results | | Interpretation |
|---|---|---|---|---|
| Response | Predictor term | ΔAIC | p value | |
| $Q$ | (1\|Gene) | −126.8 | $8.5 \times 10^{-28}$ | Imprinted genes vary in allelic bias |
| $Q$ | (1\|Dx) | 2.0 | 1.0 | Similar allelic bias for control, SCZ, AFF |
| $Q$ | (1\|Dx:Gene) | 0.4 | 0.21 | Similar gene-specific allelic bias for control, SCZ, AFF |
| $Q$ | Age | 1.3 | 0.39 | No uniform effect of age on allelic bias |
| $Q$ | (Age\|Gene) | −18.9 | $2.5 \times 10^{-5}$ | Gene-specific effect of age on allelic bias |
| $Q$ | Ancestry.1 | 0.6 | 0.24 | No uniform genetic effect on allelic bias |
| $Q$ | (Ancestry.1\|Gene) | −71.2 | $4.6 \times 10^{-16}$ | Gene specific genetic effect on allelic bias |
| $Q$ | Ancestry.3 | 1.6 | 0.54 | No uniform genetic effect on allelic bias |
| $Q$ | (Ancestry.3\|Gene) | −17.9 | $3.8 \times 10^{-5}$ | Gene-specific genetic effect on allelic bias |
| $Q$ | (1\|Gender) | 2.0 | 1.0 | No uniform male–female difference in allelic bias |
| $Q$ | (1\|Gender:Gene) | −5.7 | $5.5 \times 10^{-3}$ | Gene-specific male–female difference in allelic bias |

Predictor terms whose effect on $Q$ is modeled as random are enclosed in parentheses. Terms like (1\|Dx) or age denote effects that are uniform across imprinted genes, while terms like (1\|Dx:Gene) or (Age\|Gene) mean that the effect is specific to certain imprinted gene(s). Strongly negative ΔAIC and small $p$ value indicate significant dependence. Note that the results are based on mixed models that contained several terms besides the one tested for and shown here in the second column. For details see Methods: "Statistical models-informal overview," and beyond.

impute genotypes for SNPs not present on the Infinium SNP array using 1000 genomes data. We calibrated the imputation parameters to find a reasonable balance between the number of genes assessable for allelic bias and the number false positive calls, since the latter can arise if a SNP is incorrectly called heterozygous.

We first examined how many SNPs were heterozygous in DNA calls and had a discordant RNA call (i.e., homozygous SNP call from RNA-seq) using different imputation parameters. Known imprinted genes were excluded. We examined RNA-seq reads overlapping array-called heterozygous SNPs, which we assigned a heterozygosity score $L_{het}$ of 1, separately from RNA seq data overlapping imputed heterozygous SNPs, where the $L_{het}$ score could range from 0 to 1. After testing different thresholds, we selected an $L_{het}$ cutoff of 0.95 (i.e., imputation confidence level of 95%), and a minimal coverage of 7 reads per SNP. With these parameters,

the discordance rate (monoallelic RNA genotype in the context of a heterozygous DNA genotype) was 0.71% for array-called SNPs and 3.2% for imputed SNPs.

The higher rate of discordance for the imputed SNPs is due to imputation error. These were taken into account in two ways. First, we considered all imputed SNPs for a gene $g$ and individual $i$ jointly. Second, we excluded any individual, for which one or more SNPs supported biallelic expression.

**Quality filtering.** Two kind of data filters were applied sequentially: (1) a read count-based and (2) an individual-based. The read count-based filter removes any such pair $(i,g)$ of individual $i$ and genes $g$ for which the total read count $T_{ig} < t_{rc}$, where the read count threshold $t_{rc}$ was set to 15. The individual-based filter removes any genes $g$ (across all individuals) if read count data involving $g$ are available for less than $t_{ind}$ number of individuals, set to 25. These final filtering procedures decreased the number of genes in the data from 15,584 to $n = 5307$.

**Reference/nonreference allele test to correct for mapping bias and eQTLs.** We designed this test to distinguish imprinting from alternative causes of high read count ratio (Fig. 3): mapping bias or cis-eQTL effects. For any given gene this is a possibly compound test, since there may be multiple SNPs that are informative for the read count ratio (see Defining the read count ratio above).

For a given gene the compound null hypothesis is that the observed high read count ratio is due only to imprinting. For each informative SNP this hypothesis means that the reference and nonreference allele are associated with equal probability to the higher read count[9] (see Methods: "Defining the read count ratio to quantify allelic bias"). Thus for each SNP we assumed that the number of individuals for whom the reference allele is associated to the higher read count is binomially distributed with probability parameter 0.5. Thus, we calculated the fraction of informative SNPs for which the null hypothesis can be rejected at 0.05 significance level and used this information to decide whether the compound null hypothesis for the gene itself can be rejected.

**Test for nearly unbiased allelic expression.** The null hypothesis of this test is that the higher read count $H_{ig} = S_{ig} T_{ig}$ for gene $g$ and individual $i$ is drawn from a binomial distribution with a probability parameter $p_{ig} \approx 0.5$ suggesting nearly unbiased allelic expression. More specifically, the test was defined by the criteria

$$S_{ig} \leq 0.6 \text{ and } UCL_{ig} \leq 0.7, \tag{2}$$

where the 95% upper confidence limit $UCL_{ig}$ for the expected read count ratio $p_{ig}$ was calculated assuming that the higher read count $H_{ig} \sim \text{Binom}(p_{ig}, T_{ig})$, on the fact that binomial random variables are asymptotically (as $T_{ig} \to \infty$) normal with $\text{var}(H_{ig}) = T_{ig} p_{ig}(1 - p_{ig})$, and on the equalities $\text{var}(S_{ig}) = \text{var}(H_{ig}/T_{ig}) = \text{var}(H_{ig})/T_{ig}^2$. Therefore

$$UCL_{ig} = S_{ig} + z_{0.975}\sqrt{\frac{S_{ig}(1 - S_{ig})}{T_{ig}}}, \tag{3}$$

where $z_p$ is the $p$ quantile of the standard normal distribution.

**Data transformations.** We found transformations of the read count ratio data to be useful for fitting our statistical models (Methods: "Statistical models-informal overview", and beyond). We used either (or none) of the following two transformations:

1. The quasi-log transformation, defined as

$$\tau_Q(S_{ig}; T_{ig}) \equiv Q_{ig} = -\log\left(1 - S_{ig}\frac{T_{ig}}{T_{ig} + c}\right), \tag{4}$$

where $S_{ig}$ and $T_{ig}$ mean read count ratio and total read count for individual $i$ and gene $g$; log means natural logarithm (base $e$); $c$ is a pseudo read count set to 1 in order to avoid zero in the parenthesis since the log function is undefined at 0.

2. The rank transformation

$$\tau_R(S_{ig}; \{S_{jg}\}_j) \equiv R_{ig} = \frac{\#\{j : S_{jg} \leq S_{ig}\}}{\#j} \times 100. \tag{5}$$

Note that $j$ may equal $i$ in Eq. 5. Thus, this transformation first ranks individual $i$ among all individuals $j$ and then scales the ranks between 0 and 100.

**Statistical models-informal overview.** We modeled the dependence of read count ratio of imprinted genes jointly on all biological and technical explanatory variables (Supplementary Table 1) using several multiple regression models. Based on their structure our models can be classified into two sets of fixed and a set of mixed regression models (Supplementary Fig. 5). Furthermore our models can be also classified based on nonstructural properties (link function, error distribution, weighting, and the data transformation to read count ratio; see model classes in Supplementary Table 2).

Supplementary Fig. 5 explains that among the two fixed and the mixed structural model class the mixed one is both more powerful and robust because its random effects terms allow gene-specific parts of the model to "borrow strength from each other". The cost of the enhanced power in mixed models is the lack of estimates and confidence intervals (as well as $p$ values) for gene-specific coefficients (parameters), which the less powerful fixed models do provide (Supplementary Fig. 9). Instead of being estimated, gene-specific coefficients in mixed models therefore can only be predicted without information on confidence surrounding them (see Supplementary Fig. 10 for predicted gene-specific coefficients under a mixed model). Nonetheless, the low power and low robustness of fixed models became apparent from results like those in Supplementary Fig. 9, so we based our final inference (Table 1) on mixed modeling. Note, however, that we found an overall qualitative agreement between mixed and fixed models regarding gene-specific coefficients (compare Supplementary Fig 9 and 10).

To select the best model within the mixed structural class, we compared model fit of the nonstructural types (Supplementary Fig. 8) and found that the unlm.Q and wnlm.Q types fitted the data the best. Similar results were obtained for fixed models (Supplementary Fig. 6, 7).

**Statistical models-formal overview.** Our fixed and mixed effects multiple regression models are all generalized linear models (GLMs). GLMs in general describe a conditional distribution of a response variable $y$ given a linear predictor $\eta$ such that the distribution is from the exponential family and that $E(y|\eta) = g^{-1}(\eta)$, where $g$ is some link function. In the present context the response $y$ is the observed read count ratio that is possibly transformed to improve the model's fit to the data. We performed fitting with the lme4 and stats R packages and tested several combinations of transformations, link functions, and error distributions (Supplementary Table 2). For inference we used the best fitting combination (unlm.Q, Supplementary Table 2) as assessed by the normality and homoscedasticity of residuals (Supplementary Fig. 8, also Supplementary Fig. 6, 7) as well as by monitoring convergence.

In mixed GLMs the linear predictor $\eta = X\beta + Zb$ and in fixed GLMS $\eta = X\beta$, where $X, Z$ are design matrices containing data on explanatory variables, whereas $\beta$ and $b$ are fixed and random vectors of regression coefficients that mediate fixed and random effects, respectively (see Methods: "Detailed syntax and semantics of mixed models" and Supplementary Fig. 5 for details).

Besides the random effects term $Zb$ the key difference between the mixed and fixed models in this study is that the former describes read count ratio jointly for all imprinted genes and the latter separately for each imprinted gene. An important consequence is that our mixed models are more powerful because they can utilize information shared by all genes. Therefore we preferred mixed models for final inference and used fixed models only to guide selection among possible mixed models (Methods: "Model fitting and selection").

**Detailed syntax and semantics of mixed models.** Here we describe the detailed syntax and semantics of the normal linear mixed models combined with a quasi-log transformation $Q$ of read count ratio as this combination was found to provide the best fit (Supplementary Fig. 8). We have data on 579 individuals and 30 imprinted genes, and so the response vector is $y = (Q_{i_1 g_1}, ..., Q_{i_{579} g_1}, Q_{i_1 g_2}, ..., Q_{i_{579} g_2}, ..., Q_{i_1 g_{30}}, ..., Q_{i_{579} g_{30}})$. The model in matrix notation is

$$y = X\beta + Zb + \varepsilon \tag{6}$$

$$\varepsilon_i \overset{\text{i.i.d.}}{\sim} \mathcal{N}(0, \sigma^2), \; i = 1, ..., mn \tag{7}$$

$$b \sim \mathcal{N}(0, \Omega_b), \tag{8}$$

where the size of the covariance matrix $\Omega_b$ depends on the number of terms with random effects (the columns of $Z$). Simply put: errors and random coefficients are all normally distributed.

To clarify the semantics of Eq. 6, let us consider a simple toy model with just a few terms in the linear predictor. But before expressing it in terms of Eq. 6 it is easier to cast it in the compact "R formalism" of the stats and lme4 packages of the R language as

$$y \sim \underbrace{1 + \text{Age}}_{\text{fixed effect}} + \underbrace{(1 + \text{Age} + \text{Ancestry.1} | \text{Gene})}_{k=1}^{\text{random effects}} + \underbrace{(1 | \text{Dx} : \text{Gene})}_{k=2}. \tag{9}$$

First note that the random effect term labeled with $k = 1$ can be expanded into $(1 | \text{Gene}) + (\text{Age} | \text{Gene}) + (\text{Ancestry.1} | \text{Gene})$. The '1's mean intercept terms: one as a fixed effect and two as random effects. The first random intercept term $(1 | \text{Gene})$ expresses the gene-to-gene variability in read count ratio (compare panels in Figs 4 and 5), in other words the random effect of the gene variable. The second random intercept term $(1 | \text{Dx:Gene})$ corresponds to the interaction between psychiatric diagnosis Dx and gene; it can be interpreted as the gene-specific effect

of Dx or—equivalently—as Dx specific gene-to-gene variability. This term is not likely to be informative as Fig. 4 suggests little gene-specific effect of Dx.

We see that age appears twice: first as a fixed slope effect on $y$ and second as a gene-specific random slope effect, denoted as (Age|Gene). The random effect appears to be supported by Fig. 5 because the dependence of read count ratio on age varies substantially among genes, but the fixed effect is not supported because the negative dependence seen for several genes is balanced out by the positive dependence seen for others. The model includes another random slope effect: (Ancestry.1|Gene) with a similar interpretation as (Age|Gene) but lacks a fixed effect of Ancestry.1.

Now we are ready to write the toy model as an expanded special case of Eq. 6 as

$$y_i = \overbrace{\beta_0 + \text{Age}_i\beta_1}^{\text{fixed effects}} + \overbrace{\underbrace{b_0^{(1)} + \text{Age}_i b_1^{(1)} + \text{Ancestry.1}_i b_2^{(1)}}_{\text{Gene}_i} + \underbrace{b_0^{(2)}}_{\text{Dx}_i:\text{Gene}_i}}^{\text{random effects}} + \varepsilon_i. \quad (10)$$

As in the earlier R formalism the terms of the linear predictor are grouped into fixed and random effects. Within the latter group we have two batches of terms indicated by the $k$ superscripts on the random regression coefficients $b_j^{(k)}$. The first batch $\{b_0^{(1)}, b_1^{(1)}, b_2^{(1)}\}$ corresponds to $\{(1|\text{Gene}), (\text{Age}|\text{Gene}), (\text{Ancestry.1}|\text{Gene})\}$ in Eq. 9, the second batch contains only $b_0^{(2)}$ corresponding to (1|Dx:Gene).

Within the $k$th batch Eq. 10 contains only a single intercept coefficient $b_0^{(k)}$ and, if random slope terms are also present in the batch, only a single slope coefficient associated with the variable age or Ancestry.1. This is because only a single level of the factor gene or the composite factor Dx:Gene needs to be considered for the $i$th observation; these levels are denoted as gene$_i$ and Dx$_i$:Gene$_i$, respectively. Implicitly however, Eq. 10 contains the respective coefficients for all levels of these factors. For example, there are $n = 30$ intercept coefficients $b_0^{(1)}$ each of which corresponds to a given gene. So to generalize Eq. 10, we need $J_k$ coefficients in the $k$th batch, where $J_k$ is the product of the number of factor levels and one plus the number of random slope variables. This way we can provide the expansion of the general formula Eq. 6 using the semantics of the toy model (Eq. 9, 10) as

$$y_i = \overbrace{\sum_{j=0}^{J} x_{ij}\beta_j}^{\text{fixed effects}} + \overbrace{\sum_{k=1}^{K}\sum_{j=0}^{J_k} z_{ij}^{(k)} b_j^{(k)}}^{\text{random effects}} + \varepsilon_i. \quad (11)$$

**Model fitting and selection**. Eq. 11 describes a large set of mixed models that differ in one or more individual terms that constitute their linear predictor. From this set we aimed to select the best fitting model under the Akaike Information Criterion (AIC).

We used a heuristic search strategy in order to restrict the vast model space to a relatively small subset of plausible models. The search was started at a model whose relatively simple linear predictor was composed of terms using our prior results based on fixed effects models. The same results suggested a sequence in which further terms were progressively added to the model to test if they improve fit. Improvement was assessed by ΔAIC and the $\chi^2$-test on the degrees of freedom that correspond the evaluated term. If fit improved the term was added otherwise it was omitted. Next, further terms were tested. This iterative procedure lead to the following model.

$$Q \sim \text{RIN} + (1|\text{RNA\_batch}) + (1|\text{Institution}) + (1|\text{Institution} : \text{Individual})$$
$$+ (1|\text{Gene} : \text{Institution}) + (1|\text{Gender} : \text{Gene})$$
$$+ (\text{Age} + \text{RIN} + \text{Ancestry.1} + \text{Ancestry.3}|\text{Gene})$$

We refer to this as the "best fitting model" even thought it may represent only a local optimum in model space.

**Code availability**. All code developed by A. Gulyás-Kovács is available at: https://github.com/attilagk/monoallelic-brain-notebook The corresponding lab notebook can be browsed as a website at: https://attilagk.github.io/monoallelic-brain-notebook

**Data availability**. Data and analytical results generated through the Common-Mind Consortium are available through the CommonMind Consortium Knowledge Portal: https://doi.org/10.7303/syn2759792. Intermediate results leading to the final results published here are available from the authors at request.

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

## Acknowledgments

We thank Chaggai Rosenbluh and Ephraim Kenigsberg for valuable feedback and discussions. Data were generated as part of the CommonMind Consortium supported by funding from Takeda Pharmaceuticals Company Limited, F. Hoffman-La Roche Ltd and NIH grants R01MH085542, R01MH093725, P50MH066392, P50MH080405, R01MH097276, RO1-MH-075916, P50M096891, P50MH084053S1, R37MH057881 and R37MH057881S1, HHSN271201300031C, AG02219, AG05138, and MH06692. Brain tissue for the study was obtained from the following brain bank collections: the Mount Sinai NIH Brain and Tissue Repository, the University of Pennsylvania Alzheimers Disease Core Center, the University of Pittsburgh NeuroBioBank and Brain and Tissue Repositories and the NIMH Human Brain Collection Core. CMC Leadership: Pamela Sklar, Joseph Buxbaum (Icahn School of Medicine at Mount Sinai), Bernie Devlin, David Lewis (University of Pittsburgh), Raquel Gur, Chang-Gyu Hahn (University of Pennsylvania), Keisuke Hirai, Hiroyoshi Toyoshiba (Takeda Pharmaceuticals Company Limited), Enrico Domenici, Laurent Essioux (F. Hoffman-La Roche Ltd), Lara Mangravite, Mette Peters (Sage Bionetworks), Thomas Lehner, Barbara Lipska (NIMH).

## Author contributions

A.G.K., I.K., A.C. designed the study and interpreted results; A.G.K., I.K. implemented and performed statistical analysis; E.X. performed molecular genetics experiments; M.F., D.R. were involved with the integration of other CommonMind Consortium efforts with this project; G.H. helped with multivariate statistical modeling; R.S. supported data storage and management; G.H., R.S., M.F., D.R. gave feedback on the manuscript; A.G.K., A.C. wrote the manuscript.

## Additional information

**Competing interests:** The authors declare no competing interests.

## CommonMind Consortium

Barbara K. Lipska[11], Bernie Devlin[12,13], Chang-Gyu Hahn[14], David A. Lewis[12,13], Enrico Domenici[15], Eric Schadt[16], Hardik R. Shah[16], Jessica S. Johnson[4], Joseph D. Buxbaum[4], Lambertus L. Klei[12,13], Mette A. Peters[17], Panos Roussos[4], Raquel E. Gur[14], Solveig K. Sieberts[17], Thanneer M. Perumal[17] & Vahram Haroutunian[4]

[11]Human Brain Collection Core, National Institutes of Health, NIMH, Bethesda 20892 Maryland, USA. [12]Department of Psychiatry, University of Pittsburgh School of Medicine, Pittsburgh 15213 PA, USA. [13]Department of Human Genetics, University of Pittsburgh, Pittsburgh 15213 PA, USA. [14]Neuropsychiatric Signaling Program, Department of Psychiatry, Perelman School of Medicine, University of Pennsylvania, Philadelphia 19104 PA, USA. [15]Laboratory of Neurogenomic Biomarkers, Centre for Integrative Biology (CIBIO), University of Trento, Trento 38123, Italy. [16]Institute for Genomics and Multiscale Biology, Department of Genetics and Genomic Sciences, Icahn School of Medicine at Mount Sinai, New York 10029 NY, USA. [17]Sage Bionetworks, Seattle 98121 WA, USA

