## [Peer Review File · Nature Communications]

Reviewers' comments:

Reviewer #1 (Remarks to the Author):

In this manuscript by Gulyas-Kovacs and colleagues, the authors determine the relative contribution of specific factors to imprinting in the brain and, in particular, whether this is important in schizophrenia. These findings are of interest to the neuroscience community because many claims about the number of imprinted genes in the brain have been made as well as the potential importance of imprinting to the development of schizophrenia. Here, the authors demonstrate that while age, ancestry, and gender might be having an effect on allelic bias, there is no evidence for a relationship to schizophrenia (at least in the one brain region examined). I think these results are important for the field. Moreover, the findings are convincing and I have only minor concerns.

1. There is no abstract.
2. The authors mention that they used "pre-publication DLPFC RNA-seq data" but I assume that the authors used the published CMC dataset in reference #6? (reference #6 is not fully referenced either).
3. Removal of PCR replicates in RNA-seq data? This is necessary for SNP calling, but the replicates should be retained for expression analysis.
4. The authors claim that allelic bias dependent on age can be clearly or not clearly see in Figure 5. Is there a better way to display these data or provide other values? It is clear for ZNF331 that there is a negative dependence, but this not obvious for PEG3 as stated by the authors. These conclusions seem driven by just a few samples.
5. In Figure S3, labels for Institution, Gender, diagnosis, or ancestry are not included. Instead, there are numerical labels. Were these covariates treated as continuous variables rather than factors?

Reviewer #2 (Remarks to the Author):

Gulyás-Kovács et al. describe an approach to use RNA-seq data to identify allelic imbalances that may pinpoint imprinting. The methods are very clearly described, the analysis is rigorously conducted and the results are convincing. The manuscript is easy to follow, but still succinct. While the paper has numerous merits, it has relatively little new (positive) discovery to offer, which diminishes its importance. In particular, it does not reveal much about SCZ and might profit from putting a different spin on the story. They present a neat and well-documented method that could be applied to any RNA-seq dataset to demonstrate utility. Below I recommend some additional analyses that could be done to better "sell the method" via demonstrating true potential for new biological discovery, which it currently lacks.

Major comments:

A major limitation of the study is that is cannot relate any differential allelic bias to SCZ. In

the light of this the title is misleading and "in Schizophrenia" should be removed. Also, while I understand that the aim of the study was SCZ related, 95% of the paper tells us nothing about SCZ. One way to turn the message to something positive if the authors applied their method to other tissues to see how well genes with allelic bias are robust and which ones seem to be specific to DLPFC? Or how strongly age-dependence of imprinting can be confirmed from other tissues. These are just some suggestions given the limited set of positive results in the paper. (Negative results should be published too, obviously, but maybe not in Nat Comms.)

How robust the results are the arbitrarily chosen " $S_{\{ig\}} > 0.9$ " threshold? If 0.7 were chosen how would the top 50 genes change? Couldn't genes simply be ranked by the sum of the $S_{\{ig\}}$ scores across individuals? Similar question about the "top 50" genes, why 50, how would the top 20, top 100 behave in terms of imprinting enrichment? Wouldn't it be simpler to make the point by drawing a ROC curve (truth=known imprinted genes, $S_{\{ig\}} > .9$ fraction as predictor)? The top 50 genes then define the crucial set of 30 imprinted genes for which the mixed effect models are fitted. How robust are the model fit conclusions (e.g. about age random effect) to changes in gene selection (i.e. taking more or less imprinted genes with skewed allelic ratio?)

It is not clear to me why did the authors not test disease status as a fixed effect for one gene at a time and not only considering the 30 genes, but all ~5k genes (or at least more with variable allelic ratio) that passed the filter? According to figure 4, they tested the 30 genes one at a time (although I don't find the corresponding model described in the methods section). Why not more than 30?

The model fit descriptions on page 4 (bottom half) are far too technical for the readership of Nat Comms. Please better describe the meaning of the significant random effect components (e.g. indirect component of the age effect).

Minor comments:

1. "Of 15584 genes with RNA-seq data 5307 genes passed our filters designed to remove genes with scarce RNA-seq data reflecting low expression and/or low coverage of RNA-seq (Methods, Section 2.5)." – It seems a bit too stringent as a filter: many potential genes with allelic bias may have been thrown out at this stage. Is the method so sensitive that such harsh filters are necessary, or is it simply being overly cautious? If it is the former, it would need to be mentioned as a weakness that only a third of the genes can be tested.
2. I do not see how low power can be excluded here: "That this key result is not due to low power is indicated by the highly significant dependence of read count ratio on gene identity" Just because genes have different read count ratios, it proves that the study was well-powered to detect differences between normal and SCZ brains?
3. Where is the "nearly unbiased allelic expression" test used in the manuscript?

Thank you for your suggestions regarding improving and expanding our manuscript. As you will see, we have revised the manuscript to address the reviewers points and it is now in your Article format. This letter goes through the responses to reviewer points and associated manuscript changes.

Reviewer 1

In this manuscript by Gulyas-Kovacs and colleagues, the authors determine the relative contribution of specific factors to imprinting in the brain and, in particular, whether this is important in schizophrenia. These findings are of interest to the neuroscience community because many claims about the number of imprinted genes in the brain have been made as well as the potential importance of imprinting to the development of schizophrenia. Here, the authors demonstrate that while age, ancestry, and gender might be having an effect on allelic bias, there is no evidence for a relationship to schizophrenia (at least in the one brain region examined). I think these results are important for the field. Moreover, the findings are convincing and I have only minor concerns.

We thank this reviewer for their thoughtful review and address the specific points below.

(There were only minor comments.)

Minor comment 1. There is no abstract.

As part of reformatting to a Nature Communications Article, we now have an abstract.

Minor comment 2. The authors mention that they used “pre-publication DLPFC RNA-seq data” but I assume that the authors used the published CMC dataset in reference #6? (reference #6 is not fully referenced either).

The word "pre-publication" has been removed.

Minor comment 3. Removal of PCR replicates in RNA-seq data? This is necessary for SNP calling, but the replicates should be retained for expression analysis.

The reviewer raises an interesting point. In our previous published work (Fromer et al) we chose to remove all duplicate reads and for consistency we made the same choice in the present work. It is our thinking that this more conservative approach is warranted as the initial cDNA library was made using RNA obtained via the Ribo-Zero Magnetic Gold kit (Illumina/Epicenter Cat # MRZG12324) and random primed cDNA synthesis. Most reads with exact coincidence of start and end sites are overwhelmingly likely to indeed be PCR duplicates.

Minor comment 4. The authors claim that allelic bias dependent on age can be clearly or not clearly seen in Figure 5. Is there a better way to display these data or provide other values? It is

clear for ZNF331 that there is a negative dependence, but this not obvious for PEG3 as stated by the authors. These conclusions seem driven by just a few samples.

Yes, there are other ways to look at the data presented in Figure 5. Supporting figure Fig S4 is similar to Fig 5, with the difference that we plot the Q statistic the quasi log transformed read count ratio. Q appeared more informative than the untransformed read count ratio S on the dependence on age and other explanatory variables. Moreover, Q was used for fitting the regression models instead of S because the fit was better with Q (see figs S6 S7 S8).

Minor comment 5. In Figure S3, labels for Institution, Gender, diagnosis, or ancestry are not included. Instead, there are numerical labels. Were these covariates treated as continuous variables rather than factors?

We updated Figure S3 with the requested labels.

Reviewer 2

Gulyás-Kovács et al. describe an approach to use RNA-seq data to identify allelic imbalances that may pinpoint imprinting. The methods are very clearly described, the analysis is rigorously conducted and the results are convincing. The manuscript is easy to follow, but still succinct. While the paper has numerous merits, it has relatively little new (positive) discovery to offer, which diminishes its importance. In particular, it does not reveal much about SCZ and might profit from putting a different spin on the story. They present a neat and well-documented method that could be applied to any RNA-seq dataset to demonstrate utility. Below I recommend some additional analyses that could be done to better “sell the method” via demonstrating true potential for new biological discovery, which it currently lacks.

We thank this reviewer for their thoughtful review. We respectfully disagree with the point that we could spin the story differently. Our data and analyses allow clear conclusions regarding the extent of imprinting in human brain in neurotypic as well as schizophrenic individuals. Given the interest in imprinting in the brain as well as conjectures in the literature regarding an imprinting-schizophrenia connection, our work will be of great interest to a wide audience. We address specific points below.

Major comments:

(We have numbered the major comments.)

Major comment 1. A major limitation of the study is that is cannot relate any differential allelic bias to SCZ. In the light of this the title is misleading and “in Schizophrenia” should be removed. Also, while I understand that the aim of the study was SCZ related, 95% of the paper tells us nothing about SCZ. One way to turn the message to something positive if the authors applied their method to other tissues to see how well genes with allelic bias are robust and which ones seem to be specific to DLPFC? Or how strongly age-dependence of imprinting can be confirmed from other tissues. These are just some suggestions given the limited set of positive results in the paper. (Negative results should be published too, obviously, but maybe not in Nat Comms.)

We respectfully disagree with this point. Our view is that the title accurately reflects the data and analyses we present. We have made changes and extended the analyses addressing the other points raised by both reviewers. Regarding the suggestion to look at other tissues: in this study we chose to focus all our efforts to the DLPFC area because of manifold evidence that this brain region is relevant to SCZ, including our prior work, Fromer et al (ref #6). Had we found even weak evidence for involvement of imprinted genes' allelic bias in SCZ in the DLPFC we would have considered expanding the scope of our present study. However, given our present findings such expansion doesn't appear warranted.

Major comment 2. How robust the results are the arbitrarily chosen " $S_{ig} > 0.9$ " threshold? If 0.7 were chosen how would the top 50 genes change? Couldn't genes simply be ranked by the sum of the S_{ig} scores across individuals? Similar question about the "top 50" genes, why 50, how would the top 20, top 100 behave in terms of imprinting enrichment? Wouldn't it be simpler to make the point by drawing a ROC curve (truth=known imprinted genes, $S_{ig} > 0.9$ fraction as predictor)? The top 50 genes then define the crucial set of 30 imprinted genes for which the mixed effect models are fitted. How robust are the model fit conclusions (e.g. about age random effect) to changes in gene selection (i.e. taking more or less imprinted genes with skewed allelic ratio?)

The 0.9 threshold is not completely arbitrary because imprinted genes are known to have high allelic bias. Nonetheless we took the reviewer's advice and investigated how the gene score changes if we use threshold 0.7 instead of 0.9. We added a new supplementary figure Fig S11 that shows that the ranking with threshold 0.7 is roughly consistent with the threshold 0.9.

Regarding the query about whether genes could be ranked by the sum of the S_{ig} scores across individuals: Yes, the gene score could have been done with the sum of S_{ig} scores---or equivalently their average---across individuals for any given gene. Our approach is based on the 90th percentile instead of the average, which has two advantages. First, percentiles are less affected by outliers that may potentially emerge due to technical artifacts and the impact of sampling. Second, as we explain above the appropriate percentile can be selected (90% in this case) to reflect the biologically meaningful part of the distribution i.e. the fraction of individuals for wherein the gene in question is imprinted to a large extent. Regarding the extension of this line to asking about top 20 or top 100 in addition to top 50: Figure 3 demonstrates that below about rank 25 (lower half) most of the genes are false positives as they tend to have mapping bias and a high fraction of nearly unbiased data points (tan bars). Thus, taking the top 50 genes with respect to the gene score is sufficient and there is no need to go deeper (such as top 100).

Regarding the ROC comment: Unfortunately, ROC curves wouldn't be more useful than Fig 3. This is because ROC curves are not appropriate here since

the a priori known imprinted genes in various tissues / species / developmental stages cannot be counted on to behave in a textbook manner in the sense that they are also imprinted in the context of the adult human DLPFC. Indeed, our analyses show that they are not necessarily imprinted in that context.

Regarding the query “how robust are the model fit conclusions...” Following the reviewer’s point, we fitted the model for subsets of the genes and the results are qualitatively quite similar (but of course less significant because there are fewer genes and data points in a given subset than in the entire data set). These new results are presented in a new supplementary table (Table S3).

Major comment 3. It is not clear to me why did the authors not test disease status as a fixed effect for one gene at a time and not only considering the 30 genes, but all ~5k genes (or at least more with variable allelic ratio) that passed the filter? According to figure 4, they tested the 30 genes one at a time (although I don’t find the corresponding model described in the methods section). Why not more than 30?

We did in fact test the effect of disease status (Dx) as a fixed effect on each of the 30 imprinted genes. But we didn’t emphasize these because the fixed effects model is less robust and powerful than the mixed model. In any case the results based on fixed effects modeling were shown in Fig S9 and remain there.

We did not investigate thousands of non-imprinted genes for two reasons. First the scope of our work is imprinted genes. Second the results for imprinted genes show that disease status has at most a mild effect on read count ratio while effects of other variables (including technical ones) are substantial. Hence multivariate modeling is required to isolate the effect of disease status. But fitting multivariate regression models---especially the powerful mixed ones we used for the 30 imprinted genes---is computationally very costly. It is not feasible to do on hundreds let alone thousands of genes.

Figure 4 presents only a qualitative analysis of the potential effect of disease status on read count ratio. As mentioned earlier, quantitative analysis included both joint (mixed) modeling as well as less powerful and robust one-at-a-time (fixed effects) modeling of genes. Both kind of modeling is described in Methods sections 2.9 - 2.11. Fig S12 has been added which shows the "Distribution of read count ratio in Control Schizophrenic (SCZ) and Affective spectrum disorder (AFF) individuals for randomly selected not imprinted genes."

Major comment 4. The model fit descriptions on page 4 (bottom half) are far too technical for the readership of Nat Comms. Please better describe the meaning of the significant random effect components (e.g. indirect component of the age effect).

Table I and its legend have been carefully restructured edited and extended to simply and clearly present the hypothesis tests based on mixed models the results and the interpretation of the results.

Minor comments:

Minor comment 1. “Of 15584 genes with RNA-seq data 5307 genes passed our filters designed to remove genes with scarce RNA-seq data reflecting low expression and/or low coverage of RNA-seq (Methods, Section 2.5).” – It seems a bit too stringent as a filter: many potential genes with allelic bias may have been thrown out at this stage. Is the method so sensitive that such harsh filters are necessary, or is it simply being overly cautious? If it is the former, it would need to be mentioned as a weakness that only a third of the genes can be tested.

We share the concern of the reviewer. The problem is that we cannot make use of all reads that map to the exome but only those that map to heterozygous SNPs in the exome. This dramatically reduces the amount of data informative on allelic bias and thus after careful consideration we decided that the filtering was essential for us to be able to perform robust analyses (and avoid pitfalls that have been pointed out in some the earlier published work).

Minor comment 2. I do not see how low power can be excluded here: “That this key result is not due to low power is indicated by the highly significant dependence of read count ratio on gene identity” Just because genes have different read count ratios, it proves that the study was well-powered to detect differences between normal and SCZ brains?

The main text has been updated with the following explanation: "This key result is not due to low power. This is because in the mixed model the Gene variable (which identifies the gene that a particular data point corresponds to) is similar to the Dx variable (reporting on disease status) in that they are both categorical and are modeled as random effects. If Dx had an effect size that is comparable to the effect of Gene than that effect would be detected by our model based inference since the effect of Gene is highly significant.

Minor comment 3. Where is the “nearly unbiased allelic expression” test used in the manuscript?

We used that test to derive the black bars in Fig 3. We did actually mention this in the main text (page 4 paragraph 1) so we leave the corresponding part of the manuscript unchanged.

REVIEWERS' COMMENTS:

Reviewer #1 (Remarks to the Author):

The authors have responded to all of my concerns.

Reviewer #2 (Remarks to the Author):

The authors addressed most of my comments reassuringly. Only two points remain:

1. "Our view is that the title accurately reflects the data and analyses we present." - Normal Expression Bias of Imprinted Genes in Schizophrenia

I'm afraid I have to insist that the title, in particular the word "normal" is misleading or uninformative at best. Normal for what, compared to what? On top of it, one of the strengths of the study, the analyzed tissue, DLPFC is not even mentioned. A suggestion for a clearer title would be "No schizophrenia-specific expression bias in the dorsolateral prefrontal cortex" No need to stick to this, simply please make clearer what is meant by "normal expression bias" and potentially add DLPFC.

2. "Regarding the extension of this line to asking about top 20 or top 100 in addition to top 50: Figure 3 demonstrates that below about rank 25 (lower half) most of the genes are false positives as they tend to have mapping bias and a high fraction of nearly unbiased data points (tan bars). Thus, taking the top 50 genes with respect to the gene score is sufficient and there is no need to go deeper (such as top 100)."

The authors argue that the 0.9 is a biologically meaningful threshold. But then in the mixed model fitting, the outcome variable is the (transformed) read-count ratio, which for half of the 30 genes (Fig 3) varies very little between individuals? The authors claim that $S=0.8$ is not very different from $S=0.5$, but at the same time they try to model log/rank-transformed S for genes for which the vast majority of S is >0.9 ? Hence they try to model minute differences, such as e.g. $S=0.92$ vs $S=0.97$? I do not get the logic.

Plus they state that half of the top 50 genes are false positives. Then the question naturally emerges: why not to take only the top 25? On the other hand in the top 25 there would be even less meaningful variation in the log-transformed S (Q) values, so they would reduce power even more.

Response to Referees

Reviewer 1

The authors have responded to all of my concerns.

Reviewer 2

The authors addressed most of my comments reassuringly. Only two points remain:

1. Our view is that the title accurately reflects the data and analyses we present. - Normal Expression Bias of Imprinted Genes in Schizophrenia

I'm afraid I have to insist that the title, in particular the word normal is misleading or uninformative at best. Normal for what, compared to what? On top of it, one of the strength of the study, the analyzed tissue, DLPFC is not even mentioned. A suggestion for a clearer title would be No schizophrenia-specific expression bias in the dorsolateral prefrontal cortex No need to stick to this, simply please make clearer what is meant by normal expression bias and potentially add DLPFC.

Authors: We agree that the word "normal" might be misleading in the title.

The suggested title, however, lacks the "imprinted genes", which were the focus of our study. Therefore, we came up with a new title: "Unperturbed Expression Bias of Imprinted Genes in Schizophrenia".

2. Regarding the extension of this line to asking about top 20 or top 100 in addition to top 50: Figure 3 demonstrates that below about rank 25 (lower half) most of the genes are false positives as they tend to have mapping bias and a high fraction of nearly unbiased data points (tan bars). Thus, taking the top 50 genes with respect to the gene score is sufficient and there is no need to go deeper (such as top 100).

The authors argue that the 0.9 is a biologically meaningful threshold. But then in the mixed model fitting, the outcome variable is the (transformed) read-count ratio, which for half of the 30 genes (Fig 3) varies very little between individuals? The authors claim that $S=0.8$ is not very different from $S=0.5$, but at the same time they try to model log/rank-transformed S for genes for which the vast majority of S is >0.9 ? Hence they try to model minute differences, such as e.g. $S=0.92$ vs $S=0.97$? I do not get the logic.

Plus they state that half of the top 50 genes are false positives. Then the question naturally emerges: why not to take only the top 25? On the other hand in the top 25 there would be even less meaningful variation in the log-transformed S (Q) values, so they would reduce power even more.

Authors: We agree with the reviewer in that the effect size is relatively small. We inserted the following text as the third-to-last paragraph of the Results section: "Although some of the random effects described above are statistically significant (Table~1), their size is relatively subtle (Fig.~5). How effects translate to biological significance is an open question. Nonetheless, we focussed on genes with very strong allelic bias, in other words monoallelic expression, because that is a hallmark of imprinted genes. It is conceivable that a subtle change from, say, completely monoallelic to just nearly monoallelic expression corresponds to a qualitative change that carries some biological significance."

The authors